# Health-related quality of life and utility of maternity health states amongst post-partum Australians

**Elizabeth Martin**[1,2]*, **Olivia Fisher**[2], **Jessica Tone**[1], **Narmandakh Suldsuren**[3], **Sanjeewa Kularatna**[3], **Michael Beckmann**[1], **Yvette D. Miller**[4]

1 Mater Research Institute, University of Queensland, Brisbane, Queensland, Australia, 2 Wesley Research Institute, Brisbane, Queensland, Australia, 3 Australian Centre for Health Services Innovation, Centre for Healthcare Transformation, Queensland University of Technology, Brisbane, Queensland, Australia, 4 School of Public Health and Social Work, Queensland University of Technology, Brisbane, Queensland, Australia

* elizabeth.martin@uq.edu.au

## Abstract

**Data Availability Statement:** Our data is now available in the public repository: https://doi.org/10.48610/c933a47.

### Background

This study aimed to measure patient-reported health-related quality of life amongst post-partum women in Queensland, Australia.

### Methods

Patient-reported health-related quality of life data was prospectively collected from 134 post-partum women using the EQ-5D-5L at weekly intervals during the first six weeks following birth. Data across the five health domains of the EQ-5D-5L was converted to a single health utility value to represent overall health status. Linear mixed modelling and regression analysis were used to examine changes in utility over the first six weeks post-birth and determine associations between utility and clinical and demographic characteristics of post-partum women.

### Findings

Gestation at birth and weeks post-partum were significantly associated with utility values when considered in a multivariate linear mixed model. Mean utility values increased by 0.01 for every week increase in gestation at birth, and utility values were 0.70 at one week post-partum and increased to 0.85 at six weeks post-partum, with the largest increase occurring between one- and two-weeks post-birth. When controlling for variables that were found to predict utility values across the first six weeks post-partum, no single state of health predicted utility values at one-week post-partum.

### Conclusions

Maternity services can use our data and methods to establish norms for their own service, and researchers and maternity services can partner to conduct cost-effectiveness analysis

**Funding:** This work was funded by the Queensland University of Technology under a grant awarded to EM.

**Competing interests:** Authors EM and JT previously worked for the health service involved in this study. Employment with the health service was not related to this study and the health service did not fund this study. EM and JT were not employed at the time of data collection or manuscript preparation. MB works for the health service involved in this study. MB's employment enabled participants to be informed of the study and opportunity to participate. This does not alter our adherence to PLOS ONE policies on sharing data and materials. There are no patents, products in development or marketed products associated with this research to declare.

using our more relevant utility values than what is currently available. Time since birth and gestational age of the woman's baby should be considered when selecting post-partum health state utility values for maternity services cost-effectiveness analyses.

## Introduction

The value of maternity services for women deserves critique, given indications internationally that costs are high and health outcomes are suboptimal. Health services are experiencing increasing costs of delivering maternity care [1, 2] and private out of pocket costs for consumers are also rising [3–6]. In the United States, increasing private costs are occurring concurrently with pregnancy-related mortality ratios, which increased from 15.9 in 2012 to 17.3 per 100,000 live births in 2017, and was even higher amongst Black and American Indian, and Alaska Native women in 2017 [7]. Maternal mortality, particularly amongst ethnic minorities, is also increasing in the United Kingdom, up 24% in 2018–2020 from 2017–2019, with the six-week post-partum maternal suicide rate increasing three-fold during the same time period [8]. Suicide is the third-ranked cause of maternal death in Australia [9] and perinatal anxiety and depression prevalence is estimated between 2.6% and 40% internationally [10, 11]. Severe maternal morbidity resulting in hospital readmission is primarily due to preeclampsia, post-partum haemorrhage, renal failure and cardiac events in high income countries. There is mixed and inconclusive evidence for whether these suboptimal outcomes are due to an older maternal population and higher rates of caesarean section [12]. Hence, there is an opportunity to reassess whether maternity services can take action to improve the value of care delivered and measure outcomes in the post-partum period.

Value of maternity care can be assessed within two research paradigms: value-based health care approaches and cost-effectiveness analysis. Value-based health care approaches will compare health outcomes that matter to patients to the costs of achieving those outcomes over time, informing individual patient care and aggregate service performance [13]. In cost-effectiveness analysis, the value of one health service or intervention is determined relative to another at a population level [14]. For both value paradigms, it is not only clinical outcomes that need to be measured, but also patient-reported health-related quality of life (HRQoL) outcomes that represent the patient's perspective of their health. Clinical outcomes are indicators of a patient's diagnosis; patient-reported HRQoL outcomes are a measure of how a patient feels [13]. Whilst there is some data suggesting clinical health outcomes may be suboptimal, it is unknown for maternity services how these outcomes translate into patient-reported HRQoL outcomes which encompass a more wholistic woman-centred measure of mental and physical well-being [15].

Health-related quality of life outcomes and patient-reported outcomes are often referred to interchangeably [16], as we do in this research. Condition-specific patient-reported outcome measures (PROMs) like the Edinburgh Postnatal Depression Scale are used to assess health outcomes following engagement with health services [17]. There are many condition-specific PROMs for pregnancy, childbirth and post-partum [17–19] such as the Hyperemesis Impact of Symptoms PROM [20], Milligan's post-partum fatigue scale [21], and the International Consultation on Incontinence Questionnaire-Urinary Incontinence Short Form [22] but few capture HRQoL outcomes that might be important to women across multiple timepoints in women's maternity care journey. Generic and global HRQoL measures may also sufficiently capture outcomes that matter to women while also reducing survey burden on patients associated with responding to multiple condition-specific PROMs. There is emerging evidence that

global, generic HRQoL measures such as the PROMIS Global Short Form [23], the EQ-5D [24], or the SF-12 [25] may be used either with or without condition-specific PROMs to inform patient care and aid maternity service evaluation [26–28]. It is important to explore the use of outcomes data from global, generic HRQoL measures in maternity services as a complement or alternative to condition-specific patient-reported outcomes.

Measuring HRQoL is a critical step to improving maternity service value. Having this data from post-partum women enables improvement of maternity service value in three ways: 1) Health services can use the data to benchmark improvements in health outcomes over time [13]; 2) The relative impact of different health service improvements can be evaluated using comparable patient-reported outcomes [13]; and 3) Ideally, the data may be transformed into a form that shows how women value different states of health that they might be in at multiple time points along their maternity journey: utilities [24, 29, 30]. Health state utilities range from a value of zero, equivalent to death and one, equivalent to a state of perfect health. They are used in cost-effectiveness analysis and this method can inform the allocation of resources to improve women's post-partum health outcomes and maternity service value [14].

Despite this, there is little data on post-partum HRQoL internationally nor insights into associations between clinical or demographic variables and post-partum health; and the available studies measure HRQoL inconsistently or focus on pregnancy rather than post-partum periods [31–35]. Whilst utility value population norms for the Queensland, Australia population have been reported for age and gender categories and key preventative health indicators [36], to the best of our knowledge, there is no published data on HRQoL amongst Australian post-partum women. Publishing such data enables health systems move towards policies of value-based funding or purchasing which link funding to health service performance in terms of patient-reported outcomes, rather than volume of activity [37, 38]. To measure value in maternity services, normative data on HRQoL as a patient-reported outcome measure is critical.

Therefore, the purpose of this research was to measure patient-reported HRQoL amongst post-partum women in weekly intervals in Queensland, Australia using the EQ-5D-5L. There were three research questions: 1) What is the prevalence of each EQ-5D-5L dimension each week; 2) Are there any associations between utility and clinical or demographic characteristics of women; and 3) What are utility values for key states of health that reflect current healthcare outcomes (base-case) during the post-partum period?

## Materials and methods

### Design

This was a prospective cohort study that longitudinally collected PROMs data from post-partum women at weekly intervals over the first six weeks post-birth. This study was approved by the Queensland University of Technology (1900000297) and Mater Research (HREC/MML/51772(V3)) Human Research Ethics Committees.

### Setting

The study was conducted across three birth facilities governed by a single service in Queensland, Australia.

### Participants

The study cohort included women who birthed in one of the three study sites within the health service between 1st June and 1st December 2019. We have used the term 'women' throughout

this article. We did not collect information about the participants' gender identities and acknowledge that not all people who give birth identify as women and respect their identities and chosen terminology. Women who were <18 years of age were excluded as the EQ-5D-5L is not validated for use in this population. Women who were admitted to the intensive care unit (ICU) were also excluded. It was expected that women who did not have internet access or possess the English language skills to provide informed consent would self-exclude. Eligible patients were identified and invited to participate in the study by the maternity service. The final sample for this analysis included only women who had a live singleton birth.

The required sample size for regression analysis was determined according to Green's rule of thumb [39], where N(sample size) > 50+8m (independent variables) in the case of multiple correlations and N> 104+m in the case of testing individual predictors. Accordingly, the minimum number of participants needed for this study was 98 [N>50+(8×6)], (six variables included in Table 3) and the minimum number of samples needed for testing individual predictors was 110.

## Outcome measures

The five domains of health in the EQ-5D-5L [24, 29] were the outcome measures in this study. The EQ-5D-5L is a six-item self-reported measure of HRQoL. It asks five questions about the respondent's health at that moment, in five domains: mobility, self-care, usual activities, pain/discomfort and anxiety/depression. Each domain item asks the respondent to rate the degree of problems in that area on a five-level scale ('no problems', 'slight problems', 'moderate problems', 'severe problems', 'unable to undertake task/extreme problems'). Data collected across the five domains of health in the EQ-5D-5L was converted to a single health utility value between the value of zero (equal to death) and one (equal to perfect health) to represent overall health status using Australian country tariffs [40]. The EQ-5D-5L additionally required respondents to rate their overall condition of health on a vertical visual analogue scale between 0 ('the worst health you can imagine') and 100 ('the best health you can imagine'). The overall health rating item was not used, as the data did not answer research questions for these analyses.

Demographic and clinical information for study participants was provided by the maternity service: maternal age, body mass index (BMI), ethnicity, healthcare funding, mode of birth, parity, onset of labour, fetal presentation at birth, episiotomy, gestation at birth (weeks) and volume of blood loss during birth (mL). The Index of Relative Socio-Economic Advantage and Disadvantage (IRSAD) was calculated from women's postcode of residence at the time of birth using Australian Bureau of Statistic (ABS) 2016 census data for postal area ranking within Queensland [41] to provide a measure of socioeconomic status. Maternal length of post-partum hospital stay was calculated from the date of baby's birth and the date of mother's hospital discharge. Maternal age was categorised into 10-year age brackets and then further dichotomised to represent under 35 years of age and 35 years and older due to low frequencies of participants in the 18–24 and 44-54-year age groups.

Demographic characteristics and clinical outcomes of the whole population of women who birthed in the four-week study period was also provided by the birth facilities to assess representativeness of the respondent sample compared to the general birthing population.

## Procedure

The EQ-5D-5L was administered to participants via an online survey accessed using their mobile phones. Participants were sent a short message service (SMS) with a web link to provide informed written consent and complete the EQ-5D-5L at six time points over the first six weeks post-birth. The first SMS was sent by the birth facilities seven days after eligible participants gave birth and the five follow-up surveys were sent to consenting participants at weekly

intervals by the research team up to six weeks post-partum. Participants who missed completing the survey after providing consent at week 1 remained in the study and were sent the SMS prompt to complete the survey again at the next time point. Data, including recording of consent to participate, was securely downloaded from the survey platform in Excel files.

## Data analysis

One-sample Chi-squared statistics (categorical variables) and one-sample *t* tests (continuous variables) were used to assess equivalence between the study sample and the population of eligible women who birthed in the same birth facilities during the study time-period.

The EQ-5D-5L prevalence of each health domain by level of impairment was calculated for each of the six weeks following birth and presented as percentages with 95% confidence intervals (95% CI). The EQ-5D-5L levels were collapsed to additionally calculate the prevalence equivalent of the EQ-5D-3L. One-sample *t* tests were used to compare mean utility values over the six-week post-partum period and with Queensland, Australia age bracket population norms for females [36].

Separate linear mixed models (LMM) were conducted to examine the relationship between utility values and each clinical and demographic variable, with additional main effects of time (ordinal; week 1 (referent), week 2, week 3, week 4, week 5, week 6), and time interaction terms (clinical/demographic variable x time). Index of relative socio-economic advantage and disadvantage was not examined as a predicting variable due to low frequencies of participants in quintiles one to three and collapsing categories would not have provided meaningful independent variables for analysis. Healthcare funding (public government- or privately-funded) was considered an alternative measure of socioeconomic status. Predicting variables found to be significantly associated with utility values in initial LMMs were included as fixed effects in a final multivariate LMM. Clinical and demographic variables that were found to be significantly different from the eligible birthing population were additionally included as fixed effects to control for variations in the study sample compared to the birthing population. The final multivariate LMM was constructed to determine the association between utility values and significant clinical and demographic variables over the six-week post-partum period. Model best-fit was determined using Akaike and Bayesian Information Criteria. Random effects were specified in all models with a random intercept for participant. Restricted maximum likelihood (REML) was used with Satterwaite approximation to compute degrees of freedom. Associations were considered statistically significant at $p < .05$. Non-significant interaction terms were removed to simplify final models. Parameter estimates are only reported for significant fixed effects. Pairwise comparisons of estimated marginal means (EMMs) were conducted to further examine trends in utility values between week two to week six post-partum for the fixed effect of time. P-values for pairwise comparisons were adjusted using a Bonferroni adjustment (15 comparisons) to reduce Type 1 error associated with multiple comparisons and then considered statistically significant at $p < .05$.

Multiple regression was used to estimate the base-case utility values for key states of health at one-week post-partum, while controlling for variables found to be significantly associated with utility values in LMMs. One-week post-partum was selected because cost-effectiveness studies that might use this data are most likely to require short-term health post-partum health outcomes and we expected the largest sample size at this time point.

## Results

### Participant characteristics

One hundred and fifty-seven women consented to participate in the study. Fifteen participants did not complete the EQ-5D-5L at any time point and another four participants were missing

data on at least one explanatory variable included in the final multivariate LMM. These partici-
pants were excluded from all analyses. Participants who had a stillbirth ($n$ = 2) or multiple
birth ($n$ = 2) were additionally excluded due to insufficient frequencies to create a unique cate-
gory. The final sample included the remaining 134 participants. There were three (2.2%)
women aged 18–24; 81 (60.4%) women aged 25–35; 49 (36.6%) women aged 35–44; and one
woman (0.7%) aged 45–54 in the study sample. Compared with women who birthed in the
study sites during the same period, the sample underrepresented women aged 18 to 24 (2.2%
versus 7.8% of the birthing population; $X^2$ (1) = 102.557, $p < .001$) (Table 1). There was no dif-
ference between the study sample and the eligible birthing population when age was examined
as a dichotomous variable ($< 35$ years and $\geq 35$ years, $X^2$ (1) = 1.88, $p = .171$), and due to the
small sample sizes in each category, the dichotomous age variable was used in further univari-
ate and LMM analysis (Table 3). Women in the study sample were also more likely to be Cau-
casian ($X^2$ (1) = 15.532, $p < .001$), a private patient ($X^2$ (1) = 7.413, $p = .006$), reside in an area
within the most advantaged IRSAD quintile ($X^2$ (1) = 10.150, $p = .001$) and were at a later ges-
tation at birth ($t$(133) = 2.131, $p = .035$). There were no women in the study who identified as
Aboriginal and/or Torres Strait Islander, compared to 2.5% of women in the eligible birthing
population (See Table 1).

### EQ-5D-5L prevalence

The prevalence of no (level 1), slight (level 2), moderate (level 3), severe (level 4), and extreme
(level 5) problems in each health domain across the first six weeks following birth are pre-
sented in Table 2. Prevalence estimates with 95% CIs for the EQ-5D-3L dimensions are pre-
sented in S1 Table. Moderate or worse level problems were frequently reported in the domains
of undertaking usual activities (34.0% at one-week post-partum), pain and discomfort (19.2%
at one-week post-partum) and mobility (13.6% at one-week post-partum). The proportion of
participants reporting severe and extreme level problems was also highest in the usual activities
domain (4.0% at one-week post-partum and 3.0% at two weeks post-partum, respectively).
There was a trend of increasing prevalence of participants reporting no problems from one
week to six weeks post-partum for all health domains.

   A comparison of mean utility values over the six-week post-partum period and by age to
Queensland, Australia population norms for females is presented in S2 Table. Mean utility val-
ues at each week for the first six weeks following birth were significantly lower when compared
to females aged 25–34 years.

### Association between utility values and clinical and demographic characteristics

Separate adjusted-for-time-only LMMs indicated that maternal length of hospital stay ($F$
(143.418) = 5.677, $p = .018$), neonatal admission to SCN or NICU (($F$(138.214) = 4.307, $p =
.040$) and gestation at birth ($F$(130.657), $p = .001$) were significantly associated with utility val-
ues when examined alongside time. Maternal age ($F$(122.794) = 0.441, $p = .508$), BMI ($F$
(130.205) = 0.746, $p = .526$), ethnicity ($F$(131.288) = 0.28, $p = .868$), healthcare funding ($F$
(127.534) = 0.345), $p = .558$), mode of birth ($F$(121.498) = 2.344, $p = .076$), parity ($F$(126.106) =
0.365, $p = .547$), onset of labour ($F$(121.906) = 2.004, $p = .139$), episiotomy ($F$(123.813) = .001,
$p = .973$) and volume of blood loss ($F$(123.295) = .196, $p = .659$) were not significantly associ-
ated with utility values when examined alongside time. Time was significantly associated with
utility values in all models. There were no significant interactions between time and demo-
graphic and clinical variables.

**Table 1. Demographic and clinical characteristics of the sample compared to the eligible birthing population.**

| | Study Sample (n = 134) | Eligible Birthing Population (n = 3174) | |
|---|---|---|---|
| | *n* (%) | *n* (%) | *p*-value |
| Maternal Age | | | |
| 18–24 years | 3 (2.2) | 254 (7.8) | <0.001 |
| 25–35 years | 81 (60.4) | 1947 (59.9) | |
| 35–44 years | 49 (36.6) | 1027 (31.6) | |
| 45–54 years | 1 (0.7) | 12 (0.4) | |
| BMI | | | |
| <18.50 | 7 (5.3) | 167 (5.3) | .774 |
| 18.50–24.99 | 81 (60.4) | 1857 (59.3) | |
| 25.00–29.99 | 28 (20.9) | 626 (20.0) | |
| ≥30 | 16 (11.9) | 482 (15.4) | |
| *Missing* | *2* | *42* | |
| Ethnicity | | | |
| Caucasian/European | 102 (76.1) | 1876 (59.4) | < .001 |
| Other | 32 (23.9) | 1282 (40.6) | |
| *Missing* | *0* | *16* | |
| Aboriginal and/or Torres Strait Islander | 0 (0.0) | 78 (2.5) | N/A |
| *Missing* | *0* | *16* | |
| IRSAD score | | | |
| Quintile 1 –most disadvantaged | 6 (4.5) | 229 (7.3) | .022 |
| Quintile 2 | 3 (2.3) | 219 (6.9) | |
| Quintile 3 | 4 (3.0) | 147 (4.7) | |
| Quintile 4 | 17 (12.8) | 535 (17.0) | |
| Quintile 5 –most advantaged | 103 (77.4) | 2025 (64.2) | |
| *Reside outside QLD* | *1* | *15* | |
| *Missing* | *0* | *4* | |
| Healthcare funding | | | |
| Public | 63 (47.0) | 1859 (58.6) | .006 |
| Private | 71 (53.0) | 1315 (41.4) | |
| Mode of birth | | | |
| Unassisted vaginal birth | 65 (48.5) | 1508 (47.6) | .982 |
| Instrumental vaginal birth | 18 (13.4) | 429 (13.6) | |
| Elective caesarean birth | 35 (26.1) | 813 (25.7) | |
| Emergency caesarean birth | 16 (11.9) | 416 (13.1) | |
| *Missing* | *0* | *8* | |
| Parity | | | |
| Primiparous | 64 (47.8) | 1467 (46.2) | .725 |
| Multiparous | 70 (52.2) | 1700 (53.7) | |
| *Missing* | *0* | *7* | |
| Onset of labour | | | |
| Spontaneous | 46 (34.3) | 1218 (38.5) | .565 |
| Induced | 53 (39.6) | 1135 (35.8) | |
| No labour (elective caesarean birth) | 35 (26.1) | 813 (25.7) | |
| Missing | 0 | 8 | |
| Fetal presentation at birth | | | |
| Vertex | 125 (93.3) | 2972 (93.7) | .843 |

*(Continued)*

**Table 1.** (Continued)

| | Study Sample (*n* = 134) | Eligible Birthing Population (*n* = 3174) | |
|---|---|---|---|
| | *n* (%) | *n* (%) | *p*-value |
| Breech or Other | 9 (6.7) | 201 (6.3) | |
| *Missing* | *0* | *1* | |
| Episiotomy | 26 (19.4) | 510 (16.1) | .298 |
| Neonatal admission to SCN or NICU | 10 (7.5) | 329 (10.4) | .265 |
| *Missing* | *0* | *5* | |
| | *M (SD)* | *M (SD)* | |
| Gestation at birth (weeks) | 38.7 (2.3) | 38.3 (2.4) | .035 |
| *Missing* | *0* | *1* | |
| Maternal volume of blood loss (mL) | 401.9 (356.6) | 391.9 (337.2) | .746 |
| *Missing* | *0* | *14* | |
| Maternal length of hospital stay (days) | 3.1 (1.6) | 2.8 (1.7) | .116 |
| *Missing* | *0* | *25* | |

BMI, Body Mass Index; IRSAD, Index of Relative Socio-Economic Advantage and Disadvantage; M, Mean; N/A, not available; NICU, Neonatal Intensive Care Unit. QLD, Queensland; SCN, Special Care Nursery; SD, Standard deviation.

The model was fitted with variables selected as outlined in methods: effects of time, ethnicity (categorical; Caucasian/European (referent), other), healthcare funding (categorical; public patient (referent), private patient), neonatal admission to Special Care Nursery (SCN) or Neonatal Intensive Care Unit (NICU) (no (referent), yes), gestation at birth (continuous), and maternal length of post-partum hospital stay (continuous). This model was the best fit, having the lowest Akaike and Bayesian Information Criteria scores at -507.95 and -499.88 respectively. Estimates of fixed effects from the multivariate LMM are presented in Table 3. Time and gestation at birth were the only fixed effects that were significantly associated with utility values when considered in a multivariate model. The mean utility value increased by 0.01 for every week increase in gestation at birth ($p$ = .011). The EMMs from the final LMM for each week post-birth were: 1 week post-partum = 0.70 (Standard error [$SE$] = 0.02); 2 weeks post-partum = 0.77 ($SE$ = 0.03); 3 weeks post-partum = 0.81 ($SE$ = 0.03); 4 weeks post-partum = 0.85 ($SE$ = 0.03); 5 weeks post-partum = 0.85 ($SE$ = 0.03); 6 weeks post-partum = 0.87 ($SE$ = 0.03). When compared to one-week post-partum, utility values were significantly higher at all subsequent weeks post-partum ($p < .001$).

Pairwise comparisons between two weeks post-partum to six weeks post-partum are presented in Table 4. There were no further significant weekly changes in utility following two weeks post-partum. Between two weeks post-partum and four weeks post-partum, mean utility values increased by 0.08 (95% CI 0.02, 0.13, $p < .001$).

The mean utility value remained stable between four weeks post-partum to six weeks post-partum. The overall trend in utility values over the first six weeks following birth is displayed in Fig 1.

Mean utility values at one-week post-partum for different groups of women and states of health are presented in Table 5. When controlling for variables that were found to significantly predict utility values across the first six weeks post-partum: maternal length of hospital stay, neonatal admission to SCN or NICU and gestation at birth, no single state of health significantly predicted utility values at one-week post-partum.

**Table 2. EQ-5D-5L prevalence and 95% confidence intervals by domain and level of impairment over the six-week post-partum period.**

| | 1-week<br>% (95% CI)<br>n = 125 | 2-weeks<br>% (95% CI)<br>n = 66 | 3-weeks<br>% (95% CI)<br>n = 56 | 4-weeks<br>% (95% CI)<br>n = 67 | 5-weeks<br>% (95% CI)<br>n = 55 | 6-weeks<br>% (95% CI)<br>n = 60 |
|---|---|---|---|---|---|---|
| **Mobility** | | | | | | |
| No problems | 52.0 (43.3–60.6) | 69.7 (57.8–79.5) | 73.2 (60.4–83.0) | 84.7 (74.7–91.3) | 84.9 (73.0–92.2) | 89.8 (79.5–95.3) |
| Slight | 34.4 (26.7–43.1) | 27.3 (18.0–39.0) | 23.2 (14.1–35.8) | 15.3 (8.8–25.3) | 11.3 (5.3–22.6) | 8.5 (3.7–18.4) |
| Moderate | 12.8 (8.0–19.8) | 3.0 (0.8–10.4) | 3.6 (1.0–12.1) | – | – | 1.7 (0.3–9.0) |
| Severe | 0.8 (0.1–3.4) | – | – | – | 3.8 (1.0–12.8) | – |
| Extreme | – | – | – | – | – | – |
| **Personal Care** | | | | | | |
| No problems | 80.0 (72.1–86.1) | 88.1 (78.2–93.8) | 83.9 (72.2–91.3) | 95.6 (87.8–98.5) | 87.3 (76.0–93.7) | 91.7 (81.9–96.4) |
| Slight | 18.4 (12.6–26.1) | 10.4 (5.2–20.0) | 14.3 (7.4–25.7) | 4.4 (1.5–12.2) | 12.7 (6.3–24.0) | 6.7 (2.6–15.9) |
| Moderate | 0.8 (0.1–4.4) | 1.5 (0.2–8.0) | 1.8 (0.3–9.5) | – | – | 1.7 (0.3–8.9) |
| Severe | – | – | – | – | – | – |
| Extreme | 0.8 (0.1–4.4) | – | – | – | – | – |
| **Usual Activities** | | | | | | |
| No problems | 24.8 (18.1–33.1) | 29.9 (20.2–41.7) | 46.4 (34.0–59.3) | 50.0 (38.1–61.9) | 56.9 (44.1–68.8) | 61.7 (49.0–72.9) |
| Slight | 40.8 (32.6–49.6) | 53.7 (41.9–65.1) | 39.3 (27.6–52.4) | 42.2 (30.9–54.4) | 32.8 (22.1–45.6) | 33.3 (22.7–45.9) |
| Moderate | 28.0 (20.9–36.4) | 13.4 (7.2–23.6) | 14.3 (7.4–25.7) | 7.8 (3.4–17.0) | 6.9 (2.7–16.4) | 5.0 (1.7–13.7) |
| Severe | 4.0 (1.7–9.0) | – | – | – | 1.7 (0.3–9.1) | – |
| Extreme | 2.4 (0.8–6.8) | 3.0 (0.8–10.3) | – | – | 1.7 (0.3–9.1) | – |
| **Pain and Discomfort** | | | | | | |
| No pain or discomfort | 7.2 (3.8–13.1) | 25.8 (16.8–37.4) | 41.1 (29.2–54.1) | 48.5 (37.1–60.2) | 67.3 (54.1–78.2) | 60.0 (47.4–71.4) |
| Slight | 73.6 (65.3–80.5) | 68.2 (56.2–78.2) | 53.6 (40.7–66.0) | 47.1 (35.7–58.8) | 29.1 (18.8–42.1) | 35.0 (24.2–47.6) |
| Moderate | 16.8 (11.3–24.3) | 6.1 (2.4–14.6) | 3.6 (1.0–12.1) | 2.9 (0.8–10.1) | – | 5.0 (1.7–13.7) |
| Severe | 2.4 (0.8–6.8) | – | 1.8 (0.3–9.5) | 1.5 (0.3–7.9) | 3.6 (1.0–12.3) | – |
| Extreme | – | – | – | – | – | – |
| **Anxiety or Depression** | | | | | | |
| Not anxious or depressed | 54.4 (45.7–62.9) | 68.7 (56.8–78.5) | 66.1 (53.0–77.1) | 73.5 (62.0–82.6) | 63.6 (50.4–75.1) | 70.0 (57.5–80.1) |
| Slight | 40.0 (31.8–48.8) | 23.9 (15.3–35.3) | 28.6 (18.4–41.5) | 22.1 (13.9–33.3) | 27.3 (17.3–40.2) | 21.7 (13.1–33.6) |
| Moderate | 4.8 (2.2–10.1) | 6.0 (2.4–14.4) | 5.4 (1.8–14.6) | 4.4 (1.5–12.2) | 7.3 (2.9–17.3) | 8.3 (3.6–18.1) |
| Severe | – | 1.5 (0.3–8.0) | – | – | 1.8 (0.3–9.6) | – |
| Extreme | 0.8 (0.1–4.4) | – | – | – | – | – |

CI, Confidence interval.

Sample size varies by domain due to missing values.

–indicates cells with 0 responses for the category.

## Discussion

The aims of this study were to measure patient-reported HRQoL amongst post-partum women in Queensland, Australia and examine associations between clinical and demographic characteristics of post-partum women and health state utility values. Utility values increased significantly from 0.7 at one-week post-partum to 0.87 at six weeks post-partum, with the largest weekly change being between one and two weeks. Gestation at birth was additionally found to be associated with women's utility values over the first six weeks post-partum, with utility score increasing as gestation increases. No other health state or demographic characteristics of post-partum women were found to be associated with utility values.

**Table 3. Fixed effects and parameter estimates from the multivariate LMM predicting utility values following birth.**

| | F(*df*) | *p*-value | EMM (SE) | Estimated change in utility value [95% CI] | *p*-value |
|---|---|---|---|---|---|
| Time | 32.882 (323.367) | < .001 | | | |
| 1 week post-partum | | | 0.70 (0.02) | Ref. | |
| 2 weeks post-partum | | | 0.77 (0.03) | 0.07 [0.04–0.11] | < .001 |
| 3 weeks post-partum | | | 0.81 (0.03) | 0.12 [0.08–0.15] | < .001 |
| 4 weeks post-partum | | | 0.85 (0.03) | 0.15 [0.12–0.18] | < .001 |
| 5 weeks post-partum | | | 0.85 (0.03) | 0.15 [0.12–0.18] | < .001 |
| 6 weeks post-partum | | | 0.87 (0.03) | 0.17 [0.14–0.21] | < .001 |
| Gestation at birth | 6.655 (130.935) | .011 | | 0.01 [0.002–0.02] | .011 |
| Healthcare funding | 0.964 (133.078) | .328 | | | |
| Ethnicity | 0.001 (129.293) | .978 | | | |
| Neonatal admission to SCN or NICU | 0.135 (137.539) | .714 | | | |
| Maternal length of hospital stay (days) | 3.302 (151.177) | .071 | | | |

CI, Confidence interval; df, Degrees of freedom; EMM, Estimated Marginal Mean Utility; NICU, Neonatal Intensive Care Unit; Ref, Referent; SCN, Special Care Nursery; SE, Standard error.

This is a critical piece of work that provides data from which maternity services can benchmark improvements. Our study is novel because there is little data on post-partum HRQoL internationally nor insights into associations between clinical or demographic variables and post-partum health; and available studies measure HRQoL inconsistently. One Romanian and multiple Chinese cross-sectional studies have used a combination of measures such as the EQ-5D and the SF-12. Participants were all pregnant, rather than post-partum women without [31] and with the conditions: HIV, COVID and uterine fibroid [32–35]. In an Ethiopian study, HRQoL amongst post-partum women was measured using the SF-36, and lower HRQoL was associated with younger women aged 17–24 years, women who had never attended formal education, and women who had a caesarean section [42]. In our Australian study, we did not observe these differences across sub-groups.

**Table 4. Pairwise comparisons of estimated marginal means for time estimated from the multivariate Linear Mixed Models.**

| | Estimated mean change in utility value [95% CI] | *p*-value |
|---|---|---|
| 2 weeks post-partum | | |
| 3 weeks post-partum | 0.04 [-0.02, 0.10] | .437 |
| 4 weeks post-partum | 0.08 [0.02, 0.13] | < .001 |
| 5 weeks post-partum | 0.08 [0.02, 0.13] | < .001 |
| 6 weeks post-partum | 0.10 [0.04, 0.16] | < .001 |
| 3 weeks post-partum | | |
| 4 weeks post-partum | 0.03 [-0.02, 0.09] | 1.000 |
| 5 weeks post-partum | 0.03 [-0.03, 0.10] | 1.000 |
| 6 weeks post-partum | 0.06 [-0.03, 0.12] | .074 |
| 4 weeks post-partum | | |
| 5 weeks post-partum | 0.00 [-0.06, 0.06] | 1.000 |
| 6 weeks post-partum | 0.02 [-0.03, 0.08] | 1.000 |
| 5 weeks post-partum | | |
| 6 weeks post-partum | 0.02 [-0.04, 0.08] | 1.000 |

CI, Confidence interval.

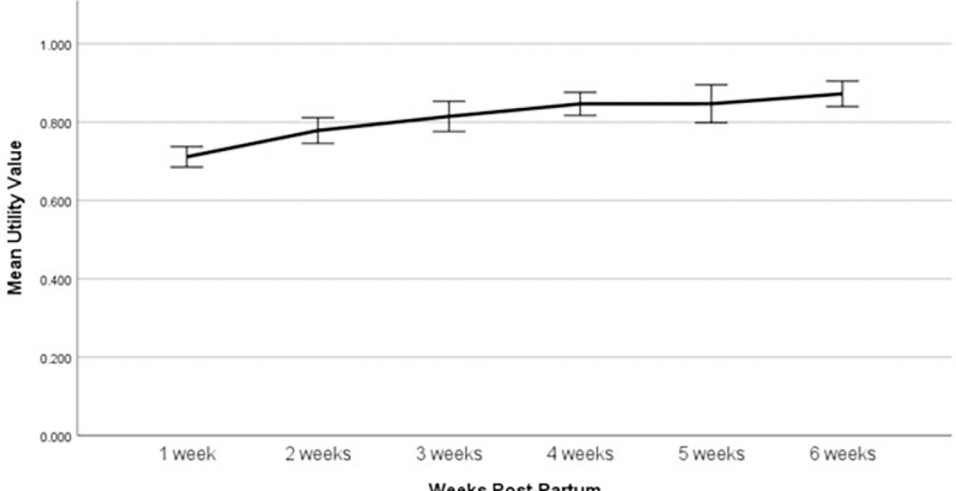

**Fig 1. Change in mean utility value over the first six-weeks following birth (Error bars denote 95% confidence intervals).**

**Table 5. Base-case utility values for key states of health at one-week post-partum ($n$ = 125).**

| Health States | Utility Value [a] M (SD) | Adjusted Utility Value[a,b] EMM (SE) | B [95% CI] | p-value |
|---|---|---|---|---|
| Maternal age | | | | |
| < 35 years | 0.70 (0.16) | 0.69 (0.03) | Ref. | Ref. |
| ≥35 years | 0.72 (0.11) | 0.72 (0.04) | 0.03 [-0.03, 0.08] | .319 |
| Mode of birth | | | | |
| Unassisted vaginal birth | 0.74 (0.14) | 0.73 (0.03) | Ref. | Ref. |
| Instrumental vaginal birth | 0.70 (0.09) | 0.68 (0.04) | -0.05 [-0.12, 0.03] | .259 |
| Elective caesarean birth | 0.67 (0.19) | 0.66 (0.04) | -0.06 [-0.13, 0.00] | .060 |
| Emergency caesarean birth | 0.72 (0.10) | 0.72 (0.04) | -0.01 [-0.10, 0.08] | .841 |
| Parity | | | | |
| Primiparous | 0.72 (0.12) | 0.70 (0.03) | Ref. | Ref. |
| Multiparous | 0.71 (0.17) | 0.70 (0.03) | -0.00 [-0.05, 0.05] | .982 |
| Onset of labour | | | | |
| Spontaneous | 0.74 (0.14) | 0.74 (0.03) | Ref. | Ref. |
| Induced | 0.71 (0.11) | 0.69 (0.03) | -0.05 [-0.11, 0.01] | .129 |
| Fetal presentation at birth | | | | |
| Vertex | 0.71 (0.14) | 0.70 (0.03) | Ref. | Ref. |
| Breech or Other | 0.68 (0.21) | 0.67 (0.07) | -0.04 [-0.16, 0.08] | .538 |
| Episiotomy | | | | |
| No episiotomy | 0.72 (0.16) | 0.72 (0.03) | Ref. | Ref. |
| Received episiotomy | 0.69 (0.11) | 0.68 (0.04) | -0.04 [-0.10, 0.02] | .214 |
| Post-partum haemorrhage (≥500ml blood loss) | | | | |
| <500ml blood loss | 0.71 (0.16) | 0.69 (0.03) | Ref. | Ref. |
| ≥500ml blood loss | 0.72 (0.11) | 0.72 (0.04) | 0.03 [-0.03, 0.09] | .320 |

CI, Confidence interval; EMM, Estimated Marginal Mean; M, Mean; Ref, Referent; SD, Standard deviation; SE, Standard error.

[a] At one-week post-partum.

[b] Adjusted for neonatal admission to SCN or NICU, gestation at birth, and maternal length of post-partum hospital stay.

Our study also contributes to maternity services being able to conduct cost-effectiveness evaluations. Internationally, many cost-effectiveness studies of maternity services or interventions do not use utility values as an outcome, reporting clinical outcomes only [43–45]. When used, utility values have been sourced from the general population [36, 46–48], or are dated [49]. We were able to compare our results to those for women of childbearing age in the general population of Queensland, Australia. Mean utility values at each week for the first six weeks following birth were significantly lower when compared Queensland, Australia population norms for females aged 25–34 years (see S2 Table). Mean utility values for women aged 25–34 years and women aged 35–44 years at one-week post-partum in our sample were also significantly lower than Queensland population norms for females in the same age brackets. These significant differences in HRQoL observed between post-partum women and population norms highlight the importance of establishing specific norms for post-partum populations to ensure accurate data informs cost-effectiveness analyses.

Future research can use our methods to prepare a larger dataset to better-inform maternity service benchmarking. Researchers and maternity services can partner to conduct cost-effectiveness analysis using our more relevant utility values than what is currently available. Time since birth and gestational age of the woman's baby should be considered when selecting post-partum health state utility values for maternity services cost-effectiveness analyses.

## Limitations

The attrition in our sample increased as time since birth increased and the cause of this is unknown. It is possible that we experienced attrition bias where women who were unwell did not continue to participate in our study. We observed sample deviation from the population across multiple characteristics and suspect that post-partum monitoring of health outcomes is not a priority for marginalised women. Due to the small sample size, our study only included women who had a live singleton birth as we did not have sufficient numbers of women who had a multiple birth or a stillbirth to create unique categories. Utility values for these women may differ from those who have had a live singleton birth. Therefore, it was not appropriate to include these women in the sample without being able to examine the utility values specific to these groups. There were also no Aboriginal and/or Torres Strait Islander women in our sample. Aboriginal and/or Torres Strait Islander women continue to experience higher rates of adverse maternal and infant health outcomes compared to non-indigenous women [9, 50, 51]. Another limitation is that our small sample was from a single Australian health service, albeit representing both public and private care. Our small sample size and exclusion of some participants, resulting in a sample that was not fully representative of the population is likely to have introduced selection bias. It should be acknowledged that other Australian maternity services may vary in the quality of care delivered and therefore the outcomes reported in our study may not be generalizable.

Future research should ideally use patient-reported quality of life outcomes data that has been collected universally through an embedded process within maternity services. A larger data set would enable clarification on whether utility values following birth differ in different service settings, between post-partum women across different culturally and linguistically diverse women, across modes of birth, age groups, and those who have multiple births or experience stillbirth.

## Conclusions

Measuring patient-reported HRQoL outcomes is a patient-centred approach to improving maternity services. It enables reorientation of maternity services to achieving outcomes that

matter to women, reducing the morbidity and mortality seen internationally, and increasing value. Our use of the EQ-5D-5L is one small step to demonstrate how maternity services can look beyond clinical indicators of wellness and measure patient-reported HRQoL. The results of this study suggest that overall quality of life is much lower than for women in the post-partum period than previous research suggested, with outcomes improving at the six-week point. Our study is an important step in the reorientation towards patient-centred care and value-based healthcare, enabling maternity services and future research to leverage both our methods and data.

## Supporting information

**S1 Table. EQ-5D-3L prevalence and 95% confidence intervals by domain and level of impairment over the six-week postpartum period.**
(DOCX)

**S2 Table. Comparison to population norms for Queensland, Australia.**
(DOCX)

## Acknowledgments

Thank you to the participants in this study who gave their time to provide data. We would also like to acknowledge the health service administrative team who sent daily text messages inviting people to participate in this study.

## Author Contributions

**Conceptualization:** Elizabeth Martin.

**Data curation:** Elizabeth Martin, Michael Beckmann.

**Formal analysis:** Elizabeth Martin, Olivia Fisher, Jessica Tone, Narmandakh Suldsuren.

**Funding acquisition:** Elizabeth Martin, Michael Beckmann, Yvette D. Miller.

**Investigation:** Elizabeth Martin.

**Methodology:** Elizabeth Martin, Sanjeewa Kularatna, Michael Beckmann.

**Writing – original draft:** Elizabeth Martin, Jessica Tone, Yvette D. Miller.

**Writing – review & editing:** Elizabeth Martin, Olivia Fisher, Jessica Tone, Narmandakh Suldsuren, Sanjeewa Kularatna, Michael Beckmann, Yvette D. Miller.

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
