## [Decision Letter · Decision Letter 0]

22 Jan 2024

PONE-D-23-28067Health-related quality of life and utility of maternity health states amongst post-partum AustraliansPLOS ONE

Dear Dr. Martin,

Thank you for submitting your manuscript to PLOS ONE. After careful consideration, we feel that it has merit but does not fully meet PLOS ONE’s publication criteria as it currently stands. Therefore, we invite you to submit a revised version of the manuscript that addresses the points raised during the review process.

We look forward to receiving your revised manuscript.

Kind regards,

Abera Mersha, MSc.

Academic Editor

PLOS ONE

Journal Requirements:

3. Thank you for stating the following in the Competing Interests section: "Authors EM and JT previously worked for the health service involved in this study but were not employed at the time of data collection. MB works for the health service involved in this study." 

4. In the online submission form, you indicated that "Due to the sensitive nature of the data, an anonymised version of the data can be requested from the corresponding author."

Reviewers' comments:

Reviewer's Responses to Questions

**Comments to the Author**

1. Is the manuscript technically sound, and do the data support the conclusions?

Reviewer #1: Yes

Reviewer #2: Yes

2. Has the statistical analysis been performed appropriately and rigorously? 

Reviewer #1: Yes

Reviewer #2: Yes

3. Have the authors made all data underlying the findings in their manuscript fully available?

Reviewer #1: No

Reviewer #2: Yes

4. Is the manuscript presented in an intelligible fashion and written in standard English?

Reviewer #1: Yes

Reviewer #2: Yes

5. Review Comments to the Author

Reviewer #1: General Comment

The study addresses a very important topic that measures whether clinical indicators such as maternal mortality ratio also translates into patient reported utilities such as quality of life (QoL). The use of standard tools such as EQ-5D-5L also ensured that their findings are comparable and replicable in other areas and therefore gives their methodology strengths. To further strengthen the paper, I have the following comments:

Specific Comments:

1. My biggest concern to this study is that I did not see whether it was adequately representative and powered to detect the outcomes that the study intended. While they reported the inclusion and exclusion criteria, it is important to show the sample size calculation to justify that the 157 that consented were adequate to study these outcomes.

2. The exclusion of 19 participants with missing data is also of concern to me as it brings in selection bias. The authors may consider including these participants in the analysis and can consider doing multiple imputation as the percentage of 12.1% missing data can be imputed to avoid per protocol analysis in line with STROBE Guidelines.

3. The differences in the sample population with the target population that gave birth at the three facilities by age group 18 – 24 but no differences when age group is stratified by below and above 35 years demonstrates that there was selection bias with the finer stratification of 18-24 which is masked when larger strata are used of below and above 35 years. I suggest you modify table 1 to reflect this and later acknowledge the sample deviation from the target population in the limitations.

4. Table 1 shows that the sample has selection bias introduced because of the sampling methodology used via texts. Women without Internet, younger age group 18 - 24 are less likely to volunteer while the wealthier women are more likely to volunteer. You can address this volunteer bias by standardization if you like or you can just acknowledge it as a limitation.

5. In table 3, they just presented adjusted utility estimates and p-values without showing unadjusted values. It is not clear how the model building was done. How were the confounders that they adjusted for selected? If the unadjusted values were presented, we can deduce that variables included were indeed confounders.

6. The final model selected in table 3 has a lot of variables that were not statistically significant. Unless they insist on author knowledge as the basis for inclusion irrespective of the evidence from the analysis, we are not sure if some of these variables are just crowding the model. I would suggest they apply Akaike and Bayesian information criteria for the full model and reduced models so that they test model best fit.

Otherwise the topic is very relevant and has potential to impact policy which mainly rely on clinical outcomes like maternal mortality without input in-put of users. For example, a mother may not die in child birth or pauperism and will be included in MMR indicator but if she is depressed or has vesico-vaginal fistula, her quality of life may not be perfect but policy makers will be glad they have reduced MMR. The methodological concerns raised may therefore help improve the paper such as the internal validity.

Reviewer #2: 1.It is recommended to update the old references (References 4, 24 and 25).

2.Be careful in using punctuation marks. In the limitations part of study, (.) is not used at the end of the sentence below:

(It should be acknowledged that other Australian maternity services may vary in the quality of care delivered and therefore the outcomes reported in our study may not be generalizable).

3.The discussion of this article can be expanded more. It is recommended to revise and make it more complete.

4.It is better to state the explanations of each table exactly before the corresponding table (review the place of inserting the explanations of tables 3 and 4)

6. PLOS authors have the option to publish the peer review history of their article (what does this mean?). If published, this will include your full peer review and any attached files.

Reviewer #1: **Yes: **Mukumbuta Nawa,

Reviewer #2: **Yes: **Raziehsadat Mousavi

---

## [Author Response · Author response to Decision Letter 0]

25 Jul 2024

We have included a file name 'Response to Reviewers' to respond to specific reviewer and editor comments.

---

## [Decision Letter · Decision Letter 1]

9 Sep 2024

Health-related quality of life and utility of maternity health states amongst post-partum Australians

PONE-D-23-28067R1

Dear Dr. Martin,

We’re pleased to inform you that your manuscript has been judged scientifically suitable for publication and will be formally accepted for publication once it meets all outstanding technical requirements.

Kind regards,

Karolina Linden, Ph.D

Academic Editor

PLOS ONE

Additional Editor Comments (optional):

Reviewers' comments:

Reviewer's Responses to Questions

**Comments to the Author**

1. If the authors have adequately addressed your comments raised in a previous round of review and you feel that this manuscript is now acceptable for publication, you may indicate that here to bypass the “Comments to the Author” section, enter your conflict of interest statement in the “Confidential to Editor” section, and submit your "Accept" recommendation.

Reviewer #1: All comments have been addressed

Reviewer #2: All comments have been addressed

2. Is the manuscript technically sound, and do the data support the conclusions?

Reviewer #1: Yes

Reviewer #2: Yes

3. Has the statistical analysis been performed appropriately and rigorously? 

Reviewer #1: Yes

Reviewer #2: Yes

4. Have the authors made all data underlying the findings in their manuscript fully available?

Reviewer #1: Yes

Reviewer #2: Yes

5. Is the manuscript presented in an intelligible fashion and written in standard English?

Reviewer #1: Yes

Reviewer #2: Yes

6. Review Comments to the Author

Reviewer #1: The authors have adequately addressed the comments I raised. Where it was not possible to change, they have included the issues as limitations in their study. This is good scientific practice and I therefore have not further comments for the authors.

Reviewer #2: Considering that similar studies have been conducted in the field, I am worried that the results of this study are not necessary for publication.

7. PLOS authors have the option to publish the peer review history of their article (what does this mean?). If published, this will include your full peer review and any attached files.

Reviewer #1: **Yes: **Dr. Mukumbuta Nawa MD, MSc, PhD

Reviewer #2: **Yes: **Raziehsadat Mousavi

---

## [Editor Report · Acceptance letter]

26 Sep 2024

PONE-D-23-28067R1 

PLOS ONE

Dear Dr. Martin, 

I'm pleased to inform you that your manuscript has been deemed suitable for publication in PLOS ONE. Congratulations! Your manuscript is now being handed over to our production team.

Kind regards, 

on behalf of

Dr. Karolina Linden 

Academic Editor

PLOS ONE